# Preparation of Bis-Thiophene Schiff Alkali–Copper Metal Complex for Metal Corrosion Inhibition

**DOI:** 10.3390/ma16083214

**Published:** 2023-04-19

**Authors:** Yafei Liu, Huixia Feng, Luyao Wang, Tiantian Yang, Jianhui Qiu

**Affiliations:** 1School of Petroleum and Chemical Engineering, Lanzhou University of Technology, Lanzhou 730050, China; liuyafei1123@163.com (Y.L.);; 2Faculty of System Science and Technology, Akita Prefectural University, Yurihonjo 0150055, Japan

**Keywords:** anticorrosion, corrosion inhibitor, thiophene Schiff base, electrochemistry

## Abstract

Due to the obvious numerous economic and technical consequences of the corrosion process, its inhibition is one of the most critical aspects of current research. A corrosion inhibitor for the bis-thiophene Schiff base copper–metal complex Cu(II)@Thy-2 was investigated here, which was synthesized via a coordination reaction with a bis-thiophene Schiff base (Thy-2) as a ligand and copper chloride (CuCl_2_-2H_2_O) as a ligand metal salt. When the corrosion inhibitor concentration was increased to 100 ppm, the self-corrosion current density *I_coor_* reached a minimum of 2.207 × 10^−5^ A/cm^2^, the charge transfer resistance reached a maximum of 932.5 Ω·cm^2^, and the corrosion inhibition efficiency reached a maximum of 95.2%, with the corrosion inhibition efficiency showing a trend of increasing first and then decreasing with concentration increase. After adding Cu(II)@Thy-2 corrosion inhibitor, a uniformly distributed dense corrosion inhibitor adsorption film formed on the surface of the Q235 metal substrate, significantly improving the corrosion profile compared to both before and after the addition of the corrosion inhibitor. Before and after the addition of corrosion inhibitor, the metal surface’s contact angle CA increased from 54.54° to 68.37°, showing that the adsorbed corrosion inhibitor film decreased the metal surface’s hydrophilicity and increased its hydrophobicity.

## 1. Introduction

Metal corrosion not only wastes a lot of energy, but also increases the risk of major safety accidents, which can cause pipeline perforations, plant explosions [1], and also weaken metal structures in bridges and buildings [2], causing them to collapse. Therefore, it has a serious impact on modern industrial development and residents’ lives and must be solved quickly and optimally. Corrosion inhibitors are often used to prevent or delay metal dissolution [3], among which, Schiff bases have received a lot of attention due to their ease of synthesis and electronic properties. Schiff bases can form complexes with almost all metal ions through azo groups. Schiff bases and their metal complexes have been widely used in medicine [4,5,6], catalysis [7,8], analytical chemistry [9,10,11], corrosion [12,13,14,15,16,17,18], photochromism [19,20,21,22], etc., and they are of great significance in the development of modern coordination chemistry and bioinorganic chemistry.

Thiophene Schiff base compounds are common organic ligands, which can quickly form chelates with transition metal ions, because there are non-bonding electrons in its -CH=N- bonds, which provide binding sites for metal ions [23]. These non-bonding electrons increase the coordination strength of Schiff bases with metal ions, resulting in thiophene Schiff base metal complexes with stable structures and excellent chemical properties [24,25]. El-Azabaw et al. [26]. quickly prepared two new Schiff base polymers (PSB1 and PSB2) with an ultra-high protective barrier function by a condensation reaction. When subjected to electrochemical tests, PSB2 inhibited acid corrosion more effectively than PSB1. In the PDP and EIS measurements, PSB2 at 100 ppm concentration exhibited 82.18% and 81.14% inhibition efficiency at 298 K, respectively. This compares to 61.85% and 67.4% inhibition efficiencies at 328 K for the same dose and measurement. Melhi et al. [27] synthesized two new cobalt(II) and chromium(III) complexes to test corrosion inhibition properties. The synthesized metal complexes were tested as corrosion inhibitors for mild steel in hydrochloric acid solutions. Electrochemical studies showed a maximum inhibition efficiency of 96.60% for Co(II)-L and 95.45% for Cr(III)-L, both complexes acting as hybrid inhibitors. As a corrosion inhibitor, the thiophene Schiff base can form a stable and dense adsorption film on metal surfaces, which is critical for corrosion protection. This corrosion inhibitor’s manufacturing process differs from previous studies. The adsorption process includes not only the corrosion inhibitor’s action on the protected metal via its adsorption groups, but also the adsorption of Cu monomers on the surface of the metal matrix caused by the substitution reaction between Cu^2+^ and Fe atoms, which is more critical than other corrosion inhibitors and can achieve better corrosion inhibition performance at low concentrations. Corrosion inhibition properties can be obtained at lower concentrations.

The unique molecular structure of Schiff bases enables their corrosion inhibitors to adsorb effectively on the surface of carbon steel, and the addition of thiophene rings can further enhance this adsorption effect. In this work, a double thiophene Schiff base (Thy-2) and copper chloride (CuCl_2_-2H_2_O) were used as ligand and ligand metal salt to prepare a double thiophene Schiff base co-metal composite corrosion inhibitor and to optimize the preparation process of the corrosion inhibitor. The corrosion inhibitor forms a denser protective film on the metal surface, providing a better physical barrier. It is worth noting that the contact angle CA on the metal surface increases from 54.54° to 68.37° before and after the addition of the corrosion inhibitor, indicating that the adsorbed film of the corrosion inhibitor reduces the hydrophilicity of the metal surface and further improves the corrosion protection performance. At the same time, we investigated the corrosion protection mechanism of the corrosion inhibitor using adsorption thermodynamic calculations and quantum chemical calculations. When used as corrosion inhibitors, the thiophene Schiff base metal complexes are expected to form more stable and denser adsorption films on metal surfaces than pure Schiff base ligands, which are of great importance for metal corrosion protection.

## 2. Materials and Methods

### 2.1. Materials

Ethanol (C_2_H_6_O, AR) and hydrochloric acid (HCl, AR) were purchased from Si-nopharm Chemical Reagent Co. (Shanghai, China). Deionized water was obtained from the laboratory. Q235 steel plates were obtained from Xiangming Hardware and Home Appliances Department, Qilihe District. Copper chloride dihydrate (CuCl_2_-2H_2_O, AR), thiophene-2-carboxaldehyde (C_5_H_4_OS, AR), DMSO-d6 (AR), and 4.6-dimethylpyrimidine (C_4_H_5_N_3_, AR) were obtained from Shanghai Aladdin Biochemical Technology Co. Acetic acid (CH_3_COOH, AR) was purchased from Sinopharm Group Chemical Reagent Co.

### 2.2. Preparation of Thy-2 Corrosion Inhibitor

In a three-necked flask, 20 mL of anhydrous ethanol and 5 mmol of 4,6-diaminopyrimidine were thoroughly mixed. We added 10 mmol of the thiophene-3-carbaldehyde solution dropwise after heating it to 80 °C. After 4 h, we added a small amount of acetic acid and stopped the reaction. The volume ratio of thiophene-3-carboxaldehyde to catalyst acetic acid was 5:1. Part of the ethanol solvent was distilled using a reduced-pressure distillation apparatus. The remaining solution was cooled and crystallized at 0 °C for 24 h to produce a pale-yellow precipitate. The resulting filter cake was dried in a vacuum oven at 40 °C for 24 h after washing and filtration with distilled water to obtain a pure 4,6-bis(thiophene-3-carbaldehyde) aminopyrimidine retardant powder labeled Thy-2.

### 2.3. Preparation of Cu(II)@Thy-2 Corrosion Inhibitor

A thiophene Schiff base (30 mL) was dissolved in ethanol, and 10 mL of copper chloride dihydrate (CuCl_2_-2H_2_O) was added. The mixture was then heated to 60 °C and stirred for 4 h, and the molar ratio of Thy-2 to CuCl_2_-2H_2_O was 10:5. The result was a brownish-brown precipitate, which was then filtered, washed with ethanol, and dried for 12 h at 60 °C in the oven. The copper–metal complex corrosion inhibitor powder with the thiophene Schiff base was acquired and identified as Cu(II)@Thy-2.

### 2.4. Characterization Tests

The structure of the experimentally created Thy-1 corrosion inhibitor was characterized using infrared absorption spectroscopy (FT-IR 850, Tianjin Guangdong Science and Technology Development Co., Ltd., Tianjin, China), X-ray diffractometer technique (XRD-6000, Shimadzu Co., Ltd., Tokyo, Japan), and X-ray photoelectron spectroscopy (Thermo Fisher K-Alpha, Waltham, MA, USA). The XRD test conditions were CuKα radiation (λ = 0.15406 nm) with a scanning range of 10–80° and a scanning speed of 0.02°/s; the FT-IR test conditions were dried, ground into powder, mixed with KBr, and compressed into tablets; the XPS test conditions were Mg Kα as the excitation source with a photoelectron energy of 1486.6 eV.

### 2.5. Electrochemical Measurements

On an electrochemical workstation (CHI660E), electrochemical tests were performed using a three-electrode system with a saturated glyceric electrode serving as the reference electrode, a platinum wire electrode serving as the auxiliary electrode, and a prepared steel sheet serving as the working electrode. Only 1 cm^2^ of the steel sheet’s surface was exposed, with the remainder being covered in sealant. Electrochemical studies, such as AC impedance spectroscopy (EIS) and kinetic potential polarization curve (Tafel) tests, were initiated after the three electrodes had been submerged in a prepared 1 M HCl solution with or without corrosion inhibitor for 30 min at room temperature. 

The EIS test frequency range was 0.01 Hz to 10 kHz, with a 10 mV AC excitation signal. For data fitting, the data from the AC impedance spectrum was analyzed using ZSimp Demo. The corrosion inhibition efficiency (η) was calculated as follows using Equation (1):(1)$$\eta =\frac{R_{ct}^{0}-R_{ct}}{R_{ct}^{}}\times 100\%$$
where *R_ct_* indicates the impedance of a Q235 specimen immersed in 1 M HCl solution for 30 min after the addition of corrosion inhibitor and $R_{ct}^{0}$ indicates the impedance of a Q235 specimen immersed in 1 M HCl solution for 30 min without the addition of corrosion inhibitor.

After the EIS test, the Tafel test was carried out with a potential range of −1 V to 0 V and a scan direction from the cathode to the anode at a rate of 1 mV/s. Using the polarization curve data, the following Equation (2) was used:(2)$$\eta =\frac{I_{coor}^{0}-I_{coor}}{I_{coor}^{0}}\times 100\%$$
where *I_coor_* indicates the self-corrosion current density of a Q235 specimen immersed in 1 M HCl solution for 30 min after the addition of corrosion inhibitor and $I_{coor}^{0}$ indicates the self-corrosion current density of a Q235 specimen immersed in 1 M HCl solution for 30 min without the addition of corrosion inhibitor.

### 2.6. Corrosion Morphology and Corrosion Product Measurement

After the electrochemical test is finished, the Q235 carbon steel sample that will be tested is chosen. The steel is then taken out of the corrosion solution, rinsed with distilled water, blown dry with cold air, and stored in a drying tower for sealed storage to prevent reoxidation.

Using scanning electron microscopy (SEM), X-ray energy spectroscopy (EDS), mapping, and contact angle tests, we examined the morphological changes of the Q235 carbon steel metal surface before and after corrosion. A contact angle tester (SZ-CAM) from Shanghai Xuanzhun Instrument Co., Ltd. (Shanghai, China) was used to measure the static contact angle. Steel was used as the specimen, and 3 μL of distilled water was dropped onto it while it was stationary. Each sample was measured three times to determine the average value.

K-Alpha (USA, Thermo Fisher) with an excitation source of Mg Kα and a photoelectron energy of 1486.6 eV was used to prepare the XPS sample, which was then used to analyze the structure of the corrosion products on the surface of Q235 carbon steel metal.

## 3. Results

### 3.1. Structural Characterization of Thy-2

Thy-2 corrosion inhibitor has 10 hydrogen atoms in its molecular structure, with two different hydrogen atoms on the pyrimidine ring, three other hydrogen atoms on each of the two thiophene rings, and the remaining two on the Schiff base group. The NMR hydrogen spectrum of Thy-2 corrosion inhibitor powder in DMSO-d6 solution is displayed in Figure 1b. As can be seen from the figure, δ = 9.10 ppm and δ = 6.41 ppm are the hydrogen proton resonance peaks of C at the α-position and the hydrogen proton resonance peak at the 5-position of the pyrimidine ring, respectively; δ = 7.67, 7.46 and 7.09 ppm are the resonance peaks of different hydrogen protons on the thiophene ring and the number of hydrogen atoms at each position is 2; δ = 8.34 ppm is the hydrogen proton resonance peak on -CH=N-, again exhibits two hydrogens. With the information provided above, it is possible to conclude that the Thy-2 molecule is a balanced combination of two thiophene aldehydes and one pyrimidine amine molecule in the presence of an ammonia-formaldehyde condensation reaction and that the hydrogen proton resonance peak of the -CH=N bond contains two hydrogens, indicating that the double thiophene Schiff base corrosion inhibitor Thy-2 molecule was successfully prepared.

### 3.2. Analysis of the Corrosion Inhibition Performance of Thy-2

The impedance spectra of Q235 carbon steel obtained using AC impedance spectroscopy EIS tests at different Thy-2 corrosion inhibitor concentrations are shown in Figure 2 above, where Figure 2a shows a Nyquist plot. In the red box, the subfigure is a partial zoom. Figure 2b shows a Bode-phase angle plot, and the electrochemical parameters in Table 1 were obtained by fitting using an equivalent circuit diagram Figure 3. The impedance spectra before and after the addition of the Thy-2 corrosion inhibitor show a semi-circular arc shape with different radii, as shown in Figure 2a, and the radius of the arc increases as the concentration of the corrosion inhibitor increases. Figure 2b shows that the impedance modulus at low frequencies increases with increasing concentration, which corresponds to the radius of the arc, and that there is a single-phase angle peak that tends to be closer to 90° as the corrosion inhibitor concentration increases. When the Thy-2 corrosion inhibitor is added to the corrosion solution, it is evident from the electrochemical parameters in Table 1 that the film capacitance and double layer capacitance tend to decrease, the deviation factor n_1_ is always 1, and n_2_ fluctuates within a specific range. Conversely, film and charge transfer resistance tends to increase as the corrosion inhibitor concentration rises. From 100 ppm to 500 ppm, the corrosion inhibitor concentration increased from 194.2 cm^2^ to 699.8 cm^2^, and the corrosion inhibition efficiency grew from 74.7% to 93.0%.

The results of the previous experiment could be attributed to the addition of a Thy-2 corrosion inhibitor, which can form a film on the metal’s surface. This film’s influence on *R_f_* is more significant due to its thickness and density. It also acts as a physical barrier, weakening the strength of charge transfer in this electrochemical system, leading to a significant increase in *R_ct_*. The film capacitance in this system exhibits pure capacitive characteristics. The corrosion inhibitor molecules gradually replace the water molecules and ions on the metal surface, extending the distance between the bilayer capacitance and lowering its capacitance value. This is indicated by the deviation factor n_1_ having a value of 1, which is evident. The phase angle peaks are simultaneously rising and convergent to 90°, which suggests that under these circumstances, a highly stable and dense adsorption coating is created on the metal surface. The adsorption type corrosion inhibitor Thy-2 can be well adsorbed on the surface of Q235 carbon steel metal, forming a layer of adsorption film with physical obstruction on its surface. When the concentration reaches 500 ppm, the film layer is dense, the metal surface coverage is the largest, and the maximum corrosion inhibition efficiency can be 93.0% at this time, playing a critical role in preventing corrosion. As a result, 93.0% of corrosion is inhibited.

### 3.3. Structural Characterization of Cu(II)@Thy-2

Figure 4a is the IR spectrum of Thy-2, Thy-2@Cu(II), and CuCl_2_·2H_2_O. For Thy-2@Cu(II), its characteristic peaks appear at 3325, 3213, 1644, 1491, 1268, 1041, 810, and 558 cm^−1^. Among them, the peak at 3325 cm^−1^ corresponds to the N-H bond on the pyrimidine ring, and the peaks at 3213 and 819 cm^−1^ are related to the aromatic C-H stretching vibration; the peak at 1491 cm^−1^ is related to the C-C and C=C bonds of the aromatic ring; and the peak at 1268 and 1041 cm^−1^ are considered to be the C-S bond adsorption peak on the thiophene ring. In addition, the peak at 1644 cm^−1^ is attributed to the stretching vibration peak of the -CH=N-bond. Compared with Thy-2, the characteristic peak has a significant blue shift, and the peak intensity has also increased. The above phenomenon is due to the fact that the lone pair of electrons in -CH=N- enters the empty orbital of metal Cu to build a stable coordination bonding [28]. The presence of this coordination bond is confirmed by the characteristic peak at 558 cm^−1^ attributed to the Cu-N stretching vibration [29]. 

The elemental composition of Thy-2@Cu(II) and the valence state of each element were characterized, and the results are shown in Figure 4. Intuitively, the signals of Cu, O, N, C, and S elements all appear in the XPS spectrum of Thy-2@Cu(II) (Figure 4b). For C1s, its high-resolution spectrum can be fitted by two peaks, which are the peak at 284.71 eV corresponding to the C-C/C=C bond and the peak at 287.13 eV corresponding to the C=S/C=N bond (Figure 4c). The high-resolution spectrum of N1s can also be fitted by two peaks (Figure 1d), which are attributed to the -N- bond in the pyrimidine ring (399.39 eV) and the -N=CH bond in the Schiff base (399.81 eV), respectively. For Cu2p, peaks matching CuCl_2_ appear at 932.27 eV and 952.04 eV. It should also be noted that there is a fitting peak corresponding to the coordination bond between metal Cu and Schiff base at copper–metal complex corrosion inhibitor was successfully prepared. 

### 3.4. Analysis of the Corrosion Inhibition Performance of Cu(II)@Thy-2

We investigated the effect of the Cu(II)@Thy-2 corrosion inhibitor solution at different concentrations on the Tafel curve of Q235 carbon steel in 1 M HCl solution, the results are shown in Figure 5, and the pertinent calculated parameters are listed in Table 2. Tafel curve testing was used to assess the performance of the Cu(II)@Thy-2 complex corrosion inhibitor. Figure 5 illustrates how the cathode Tafel curve’s shape altered somewhat before and after the injection of the corrosion inhibitor. When the corrosion inhibitor concentration was increased from 20 ppm to 100 ppm, the self-corrosion current density *I_coor_* decreased from 3.282 × 10^−5^ to 2.207 × 10^−5^ A/cm^2^, and the corresponding corrosion inhibition efficiency η increased from 93.6% to 95.7%, as can be seen when combined with the electrochemical parameters in Table 2; The *I_coor_* rises, and η correspondingly falls as the corrosion inhibitor concentration increases. The self-corrosion potential fluctuates within a narrow range (−0.445 V) in response to changes in corrosion inhibitor concentration, and the slope of the cathodic polarization curve β_c_ remains essentially constant (7.865 mV·dec^−1^). In contrast, the slope of the anodic polarization curve β_a_ experiences a slight increase following the addition of corrosion inhibitor but does not continue to exhibit a regular trend as corrosion inhibitor concentration increases.

According to the previous experimental data analysis results, the anodic metal dissolution reaction and cathodic hydrogen precipitation reaction are inhibited to varying degrees by the addition of Cu(II)@Thy-2 compound corrosion inhibitor, with the anodic reaction being inhibited to a greater extent and resulting in a significant increase in β_a_ before and after the addition of the corrosion inhibitor. As such, the corrosion inhibitor is the primary anodic protection of the hybrid. We can accurately predict the corrosion rate of metals by extrapolating the cathodic polarization curve’s slope [30]. When the corrosion inhibitor concentration rises to 100 ppm, the active corrosion site is covered by sizeable metal–organic compound corrosion inhibitor molecules. The thickness and coverage of the corrosion inhibitor adsorption film generated on the metal surface are rising. As a result, the metal substrate is effectively protected. Corrosion inhibition performance decreases as corrosion inhibitor concentration rises, possibly because the corrosion inhibitor contains free metal Cu^2+^, and the Fe atoms in carbon steel substrate are susceptible to redox reactions. As a result, Fe atoms are oxidized into Fe^2+^ and Fe^2+^, but this process also encourages metal dissolution reactions on the metal surface, leading to a faster corrosion rate.

Figure 5, where Figure 5b is the AC impedance spectrum and Figure 5c is the Bode-phase angle diagram, displays the EIS impedance spectrum study of Q235 carbon steel in solutions containing various concentrations of Cu(II)@Thy-2 complex corrosion inhibitor. Table 3 lists the EIS fitting parameters derived using the fitting circuit, and Figure 6 displays the fitted equivalent circuit. The shape of the impedance spectra for the blank group is similar to the shape of the impedance spectra shown in Figure 5b at various concentrations, demonstrating that adding a corrosion inhibitor did not affect the corrosion mechanism of Q235 carbon steel in 1 M HCl solution. The radius of the arc R_p_ exhibits a growing and subsequently decreasing pattern *R_ct_* when the corrosion inhibitor concentration is not assumed to increase, which is consistent with the trend of the fitted parameter values in Table 3. As can be seen from Figure 5b, the impedance modulus at low frequencies is also synchronous with the trend of R_p_, and there is one and only one phase angle peak in the phase angle curve before and after the addition of the corrosion inhibitor, which is closer to 90° compared to the blank group. Combined with the fitted parameters in Table 3, it can be seen that after the addition of the corrosion inhibitor, the constant phase element CPE correlation value Y_0_ and the deviation factor n decreased. At the same time, the solution impedance R_s_ and the charge transfer resistance *R_ct_* increased, with a more significant increase in *R_ct_*. As the corrosion inhibitor concentration increased from 20 ppm to 100 ppm, *R_ct_* increased from 607.8 to 932.5 Ω·cm^2^, with a corresponding rise in η from 91.9% to 94.7%; However, as the corrosion inhibitor concentration continues to increase, *R_ct_* begins to decrease and η likewise begins to decrease.

The molecular structure of the Cu(II)@Thy-2 corrosion inhibitor contains more adsorption groups, which can form a good adsorption effect on the carbon steel metal surface. The final adsorption film can include a pseudo-capacitance structure with the metal substrate, represented by the constant phase element CPE and the double layer capacitance C_dl_ in the fitted circuit. This is one possible explanation for the analysis of the aforementioned experimental data. Following the addition of the corrosion inhibitor, the water molecules that were initially adsorbed on the metal surface are replaced by the corrosion inhibitor macromolecules, and the capacitance spacing increases. This causes a decrease in the magnitude of the capacitance value that represents the CPE. In contrast, an increase in the roughness of the metal surface causes a reduction in the n value, indicating that the CPE’s pure capacitance characteristics are weakening. As the concentration of the corrosion inhibitor is increased to 100 ppm, the adsorbed film layer progressively forms, and the properties of the bilayer capacitance start to appear, along with a gradual rise in the capacitance value C_dl_; the corrosion inhibitor macromolecule blocks the charge transfer channel during the anodic metal dissolution reaction, causing the response active site to vanish. As a result, the *R_ct_* value rises, and the corrosion inhibition effect is continuously improved. However, as the corrosion inhibitor concentration increases, the *R_ct_* value starts to fall, likely because too much Cu^2+^ reacts with Fe in the redox reaction, reducing the Fe atoms.

In conclusion, it was discovered that the corrosion inhibitor concentration significantly impacts the corrosion inhibition effect of Q235 carbon steel in 1 M HCl solution and that when the corrosion inhibitor concentration reaches 100 ppm, the corrosion inhibition efficiency can reach a maximum. The corrosion inhibition performance of Cu(II)@Thy-2 complex corrosion inhibitor was evaluated using the Tafel curve and EIS impedance spectroscopy test method in electrochemistry.

The surface morphology and element content of the metal were analyzed after corrosion (Figure 5d and Table 4). The unevenness of the metal surface without the addition of corrosion inhibitor indicates that the metal has been severely corroded. After adding the corrosion inhibitor, although the metal surface still shows unevenness, the degree has been significantly improved, suggesting that the corrosion inhibition effect of the corrosion inhibitor is obvious. When the concentration of corrosion inhibitor is 400 ppm, the protected metal surface appears smoother and denser, and the corrosion inhibition efficiency will be further improved in theory, but this is not the case. This is because when the concentration of corrosion inhibitor is greater than 100 ppm, the decisive factor affecting the corrosion inhibition performance of Cu (II)@Thy-2 is no longer the coverage of its adsorption film on the metal surface. 

EDS analysis showed that the contents of N and Cu elements increased significantly due to the addition of corrosion inhibitors, which was based on the formation of adsorption films on metal surfaces by corrosion inhibitors. Comparing Figure 5c,d, their Ncontent is basically the same, but the Cu content of the latter is significantly increased. This may be attributed to the replacement reaction between the Cu ions of the corrosion inhibitor and the metal matrix when the concentration of the corrosion inhibitor was 400 ppm, and the Cu ions were reduced to simple Cu and adsorbed on the metal surface.

In order to further reveal the corrosion inhibition effect of the corrosion inhibitor on the metal, the surface contact angle test of the metal after being corroded under different conditions was carried out (Figure 5e). The results show that the CA before and after adding the corrosion inhibitor is smaller and larger than that of the original metal, respectively. This is because the passivation layer produced by the corrosion of Q235 carbon steel by acid solution improves the hydrophilicity of the metal surface, thus making it easier for the corrosive medium to contact the metal surface. However, the corrosion inhibitor forms an adsorption film on the surface of carbon steel to increase its hydrophobicity, and the physical barrier effect of the adsorption film can bring excellent corrosion inhibition effect.

### 3.5. Analysis of Corrosion Products and Revelation of Corrosion Inhibition Mechanisms

The XPS analysis of the surface products produced by the corrosion of Q235 carbon steel in hydrochloric acid solution containing corrosion inhibitor is shown in Figure 7. The signals of C, N, O, Cu, Fe and Cl elements all appear in the total spectrum (Figure 7a). Figure 3b depicted the XPS spectral analysis of the C1s fit, which showed three fitted peaks: the fitted peak at 283.62 eV, which can be attributed to C=C, C-C, and C-H bonds; the fitted peak at 285.01 eV, which can be attributed to a C=S/N bond in a heterocyclic ring; and the fitted peak at 287.34 eV, which can be represented as a Schiff base group-CH=N bond.

For N1s (Figure 7c), its peak splitting result is consistent with that of corrosion inhibitor (Figure 4d). For Cu2p (Figure 7d), different from corrosion inhibitor, its peak splitting situation is more complicated. It can be fitted into five peaks, which are the fitting peaks at 931.41 and 951.21 eV corresponding to CuCl_2_, representing the fitting peak at 933.63 eV of the coordination bond of the corrosion inhibitor [31], the fitting peaks at 931.48 eV of elemental.

Cu (based on the replacement reaction between Cu ions and Fe elements [32,33]). The appearance of elemental Cu well explains that the continuous increase in the corrosion inhibitor concentration cannot promote the improvement of its corrosion inhibition, on the contrary, it will decrease. The Fe2p spectra (Figure 7e) can be fitted into four peaks centered at 709.90, 713.29, 717.88, and 723.21 eV, attributed to Fe^3+^ of Fe_2_O_3_/FeOOH, FeCl_3_, CuFeO [34], and Fe ion chelate, respectively.

We investigated the binding ability of Thy-2 at different concentrations to Q235 carbon steel surface (Figure 7f–h). The adsorption equilibrium constants and standard free energy of adsorption parameters for the various test techniques are shown in Table 5. The results suggest that the Langmuir adsorption model is more suitable for the adsorption process of corrosion inhibitor molecules on the carbon steel surface in 1 M HCl solution. The Gads values obtained based on Tafel and EIS curves are all negative, indicating that the adsorption process of corrosion inhibitors on the carbon steel surface is spontaneous. Meanwhile, the |${∆G}_{ads}^{0}$| are all greater than 40 kJ/mol, which indicates that the adsorption process of corrosion inhibitor on carbon steel surface is chemical adsorption. 

The two main sources of this chemical adsorption are the adsorption of elemental Cu on the metal surface, and the combination of unbonded iron ions on the surface of the metal substrate and the corrosion inhibitor to produce chelates. It is based on these two chemical adsorption processes that the corrosion inhibitor can form a dense and stable protective film on the surface of the metal substrate, thus exhibiting excellent corrosion inhibition performance.

## 4. Conclusions

A double thiophene Schiff base copper–metal complex Cu(II)@Thy-2 corrosion inhibitor was investigated here, which was synthesized after a coordination reaction utilizing double thiophene Schiff base (Thy-2) as ligand and copper chloride (CuCl_2_-2H_2_O) as ligand metal salt. When the corrosion inhibitor concentration was increased to 100 ppm, the self-corrosion current density *I_coor_* reached a minimum of 2.207 × 10^−5^ A/cm^2^, the charge transfer resistance reached a maximum of 932.5 Ω·cm^2^, and the corrosion inhibition efficiency reached a maximum of 95.2%, and the corrosion inhibition efficiency showed a trend of increasing and then decreasing with increasing concentration. The corrosion morphology analysis results revealed that the metal substrate surface of Q235 after the addition of Cu(II)@Thy-2 corrosion inhibitor had a layer of uniformly distributed and dense corrosion inhibitor adsorption film, which had significantly improved when compared to the corrosion morphology before and after the addition of the corrosion inhibitor; the contact angle CA of the metal surface before and after the addition of the corrosion inhibitor increased from 54.54° to 68.37°, showing that the corrosion inhibitor adsorbed film reduced the metal surface’s hydrophilicity and increased its hydrophobicity. Finally, when used as corrosion inhibitors, thiophene Schiff alkali–metal complexes can form a more stable and dense adsorption film layer on the metal surface, which is critical for metal corrosion protection.

## Figures and Tables

**Figure 1 materials-16-03214-f001:**
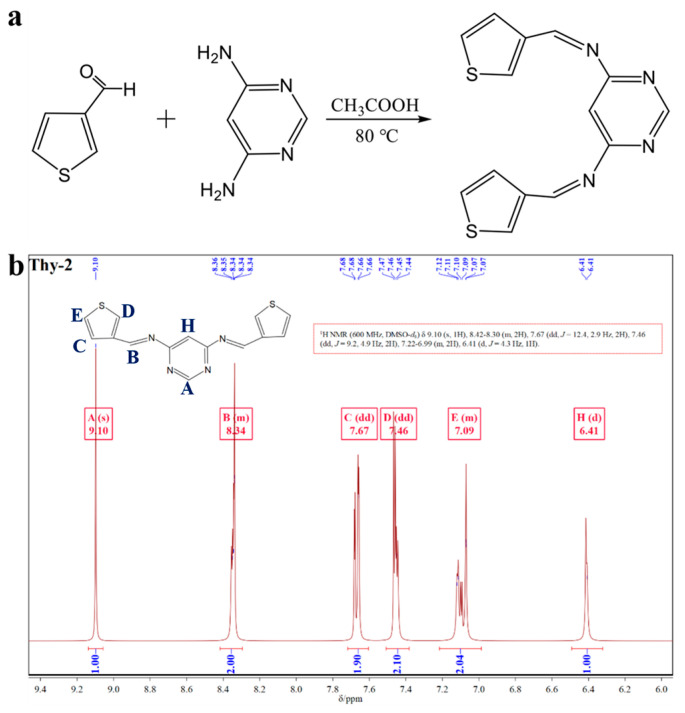
(**a**) Synthesis route of Thy-2 corrosion inhibitor; (**b**) ^1^H-NMR spectrum of Thy-2 inhibitor molecule.

**Figure 2 materials-16-03214-f002:**
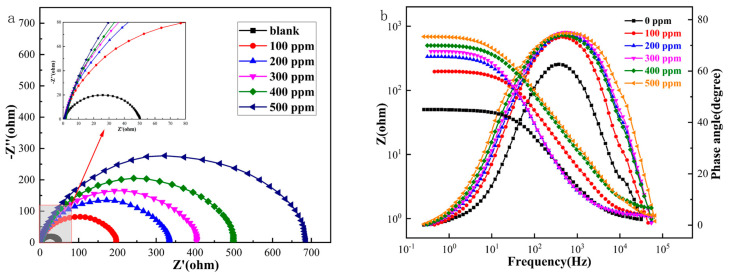
AC impedance spectrum of Q235 carbon steel in solution containing different.

**Figure 3 materials-16-03214-f003:**
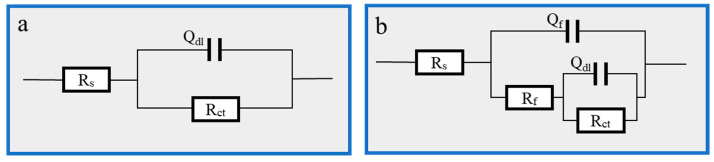
EIS fitting equivalent circuit: (**a**) 1 M HCl solution; (**b**) Thy-1 corrosion inhibitor is added.

**Figure 4 materials-16-03214-f004:**
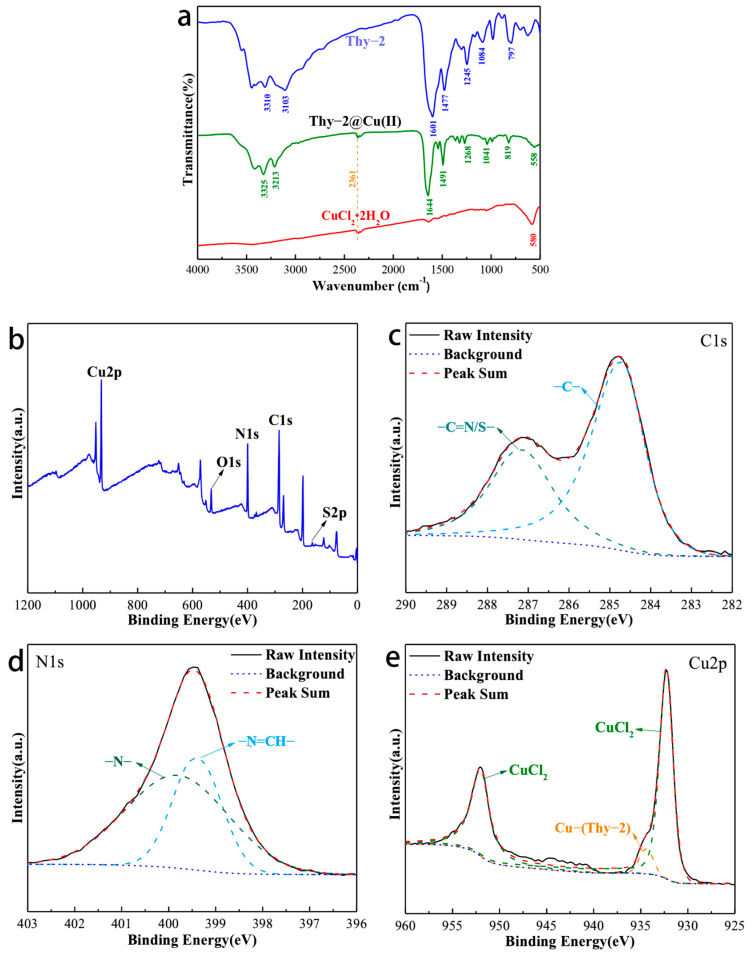
(**a**) Infrared spectrum of Thy-2, Cu(II)@Thy-2 and CuCl2·2H_2_O; (**b**) XPS spectrum of Cu(II)@Thy-2 inhibitor; (**c**) Survey scan spectra; (**d**) Narrow scan spectra of C1s; (**e**) N1s; d) Cu2p.

**Figure 5 materials-16-03214-f005:**
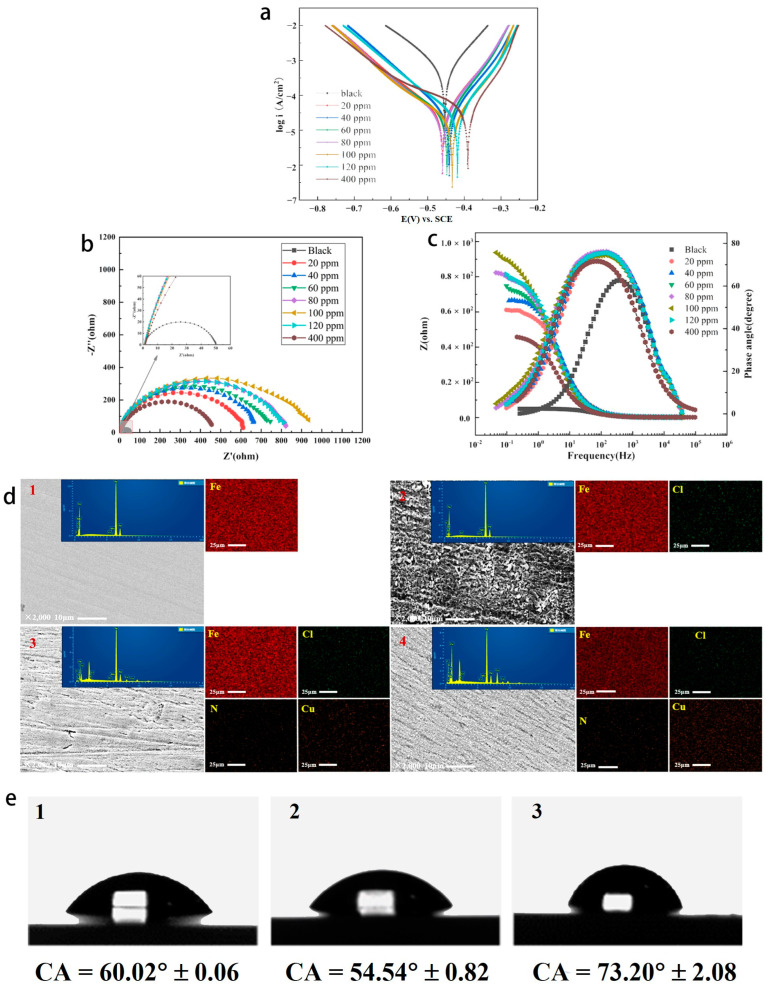
(**a**) Tafel curves of Q235 carbon steel in solutions containing different concentrations of Cu(Ⅱ)@Thy-2; Tafel curves of Q235 carbon steel in solutions containing different concentrations of Cu(Ⅱ)@Thy-2; (**b**) Nyquist spectrum; (**c**) Bode-phase angle diagram; (**d**) SEM(10 μm), EDS(25 μm) and mapping of Q235 carbon steel surface: (1) Bare steel sheet; (2) 1 M HCl solution; (3) 100 ppm Cu(Ⅱ)@Thy-2; (4) 400 ppm Cu(Ⅱ)@Thy-2; (**e**) CA analysis of Q235 carbon steel metal surface: (1) Bare steel sheet; (2) 1 M HCl solution; (3) 100 ppm Cu(Ⅱ)@Thy-2 solution.

**Figure 6 materials-16-03214-f006:**
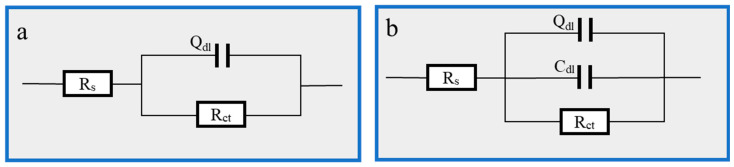
EIS fitting equivalent circuit: (**a**) 1 M HCl; (**b**) Cu(II)@Thy-2 corrosion inhibitor added.

**Figure 7 materials-16-03214-f007:**
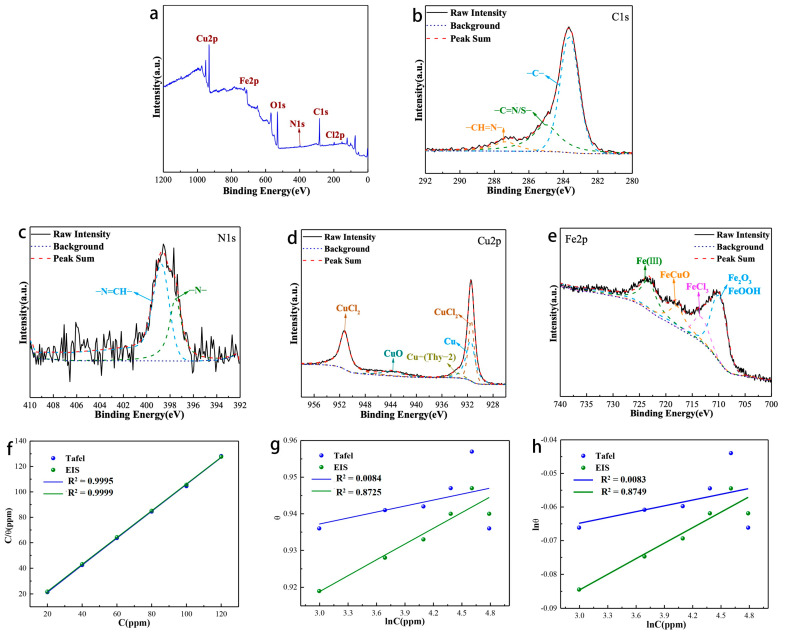
XPS analysis of corrosion products: (**a**) survey scan spectra; (**b**) narrow scan spectra of C1s, (**c**) N1s, (**d**) Cu2p, (**e**) Fe2p; (**f**–**h**) Adsorption isotherm model of Thy-1 corrosion inhibitor on Q235 carbon steel surface.

**Table 1 materials-16-03214-t001:** EIS fitting parameters of Q235 carbon steel in solution containing different concentrations of Thy-2.

*C*/(ppm)	$C_{f}(n_{1})$ $(\mu F\cdot {\mathrm{cm}}^{-2})$	$C_{dl}(n_{2})$ $(\mu F\cdot {\mathrm{cm}}^{-2})$	$R_{f}$ $\left(\Omega \cdot {\mathrm{cm}}^{2}\right)$	$R_{ct}$ $\left(\Omega \cdot {\mathrm{cm}}^{2}\right)$	$R_{p}$ $\left(\Omega \cdot {\mathrm{cm}}^{2}\right)$	η/(%)
0	-	85.23 (0.8702)	-	49.1	49.1	-
100	14.73 (1.0000)	10.13 (0.7579)	11.3	182.9	194.2	74.7
200	13.87 (1.0000)	11.48 (0.7424)	12.01	324.6	336.6	85.4
300	12.80 (1.0000)	10.39 (0.7613)	13.50	395.9	409.4	88.0
400	10.79 (1.0000)	11.02 (0.7373)	17.18	499.2	516.4	90.5
500	9.128 (1.0000)	10.63 (0.7401)	15.76	684.0	699.8	93.0

**Table 2 materials-16-03214-t002:** Tafel curves parameters of Q235 carbon steel in solutions containing different concentrations of Cu(II)@Thy-2.

*C*/(ppm)	$E_{coor}/(V)$	$I_{coor}/(A\cdot {\mathrm{cm}}^{-2})$	$\beta_{\alpha }/(\mathrm{mV}\cdot {\mathrm{dec}}^{-1})$	$\beta_{c}/(\mathrm{mV}\cdot {\mathrm{dec}}^{-1})$	η/(%)
0	−0.453	5.101 × 10^−4^	11.668	8.355	-
20	−0.460	3.282 × 10^−5^	12.749	8.154	93.6
40	−0.441	3.009 × 10^−5^	12.839	9.484	94.1
60	−0.448	2.974 × 10^−5^	13.667	7.044	94.2
80	−0.459	2.807 × 10^−5^	13.225	7.897	94.7
100	−0.433	2.207 × 10^−5^	13.474	6.476	95.7
120	−0.419	3.256 × 10^−5^	13.579	7.646	93.6

**Table 3 materials-16-03214-t003:** EIS curves parameters of Q235 carbon steel in solutions containing different concentrations of Cu(II)@Thy-2.

C/(ppm)	$R_{s}/(\Omega \cdot {\mathrm{cm}}^{-2})$	$C_{\mathrm{dl}}/(\mu F\cdot {\mathrm{cm}}^{-2})$	$Y_{0}/(s^{n}\cdot \Omega \cdot {\mathrm{cm}}^{-2})$	n(0<n<1)	$R_{ct}/(\Omega \cdot {\mathrm{cm}}^{-2})$	η/(%)
0	0.9815	-	1.735 × 10^−4^	0.8702	49.11	-
20	1.427	12.86	0.8934 × 10^−5^	0.8571	607.8	91.9
40	1.301	19.22	0.8867 × 10^−4^	0.8303	679.3	92.8
60	1.241	25.07	1.042 × 10^−4^	0.8051	734.7	93.3
80	1.135	32.75	1.215 × 10^−4^	0.7800	817.6	94.0
100	1.226	32.04	1.550 × 10^−4^	0.7356	932.5	94.7
120	1.295	23.97	1.086 × 10^−4^	0.8021	814.8	94.0

**Table 4 materials-16-03214-t004:** Atomic mass percentage of each element in EDS analysis.

Surface Analysis	Kandungan Unsur (wt%)								
	Fe	C	O	Cr	Mn	Cl	S	N	Cu
Setelah dipoles	90.36	8.06	1.28	0.10	0.20	-	-	-	-
1 M HCl	73.85	22.33	3.51	0.05	0.16	0.10	-	-	-
100 ppm Cu(II)@Thy-2	66.48	23.98	0.83	0.01	0.13	0.09	-	4.63	3.85
400 ppm	52.82	23.71	1.24	-	0.09	0.02	-	3.63	18.39

**Table 5 materials-16-03214-t005:** Adsorption thermodynamic parameters of Q235 carbon steel in 1 M HCl calculated under different test conditions.

	$K_{ads}/(M^{-1})$	$\Delta G_{ads}^{0}/(\mathrm{kJ}\cdot {\mathrm{mol}}^{-1})$
Tafel	227,560	−51.65
EIS	883,947	−49.03

## Data Availability

The data used to support the findings of this study are included within the article.

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
