# Peer review of "Preparation of Bis-Thiophene Schiff Alkali–Copper Metal Complex for Metal Corrosion Inhibition"

_materials, 2023, doi:10.3390/ma16083214_

Round 1
Reviewer 1 Report
caption 1. Introduction is missed
re-write : paragraph 2.1.Preparation of Thy-2 corrosion inhibitor
a scheme with the reaction for the obtention of Thy-2 would clear the structure of the everywhere mentioned Schiff base.
the structure of Thy-2 should be at least confirmed by 1H-NMR, I think IR is not enough to see the purity of the compound used.
Author Response
1、caption 1. Introduction is missed
We were really sorry for our careless mistakes. Thank you for your reminder. We have made corrections.
2、re-write : paragraph 2.1.Preparation of Thy-2 corrosion inhibitor
We sincerely thank the reviewer for careful reading. As suggested by the reviewer. We have re-written.
3、a scheme with the reaction for the obtention of Thy-2 would clear the structure of the everywhere mentioned Schiff base.
We have added a roadmap for the synthesis of Thy-2 to the manuscript.
4、the structure of Thy-2 should be at least confirmed by 1H-NMR, I think IR is not enough to see the purity of the compound used.
As suggested by the reviewer. We have added 1H-NMR of Thy-2 to the manuscript.
Reviewer 2 Report
- All the comments are listed in the paper.
- Electrochemical measurements results are preliminary type and the discussion is scarce or non-existent. Can you explain these results in more detail?
- compare the corrosion current density estimated from the Tafel polarization plots with those that can be calculated from the polarization resistance values obtained from EIS data
-EIS results must go into the main text and along with the experimental results, also show the fitted EIS results
- It would be necessary to give, the scattering (error bars) of all fitted parameters to evaluate the accuracy of the used model.
- the DFT calculations are strongly recommended
-Update with new conclusions.
-Checks all references and correct by the corresponding style of writing references, (journal, journal abbreviation, first and last pages, commas, sources from Web pages are missing a document access date, etc....).

Author Response
1、All the comments are listed in the paper.
We think this is an excellent suggestion. We have added relevant content to the manuscript.
2、 Electrochemical measurements results are preliminary type and the discussion is scarce or non-existent. Can you explain these results in more detail?
We have re-written this part according to the Reviewer’s suggestion.
3、compare the corrosion current density estimated from the Tafel polarization plots with those that can be calculated from the polarization resistance values obtained from EIS data.
We sincerely appreciate the valuable comments. We have carefully checked the literature on corrosion inhibitors and have not found the relevant comparisons you mention. If you have similar literature on your side, we would be grateful if you could provide us with the relevant information.
4、EIS results must go into the main text and along with the experimental results, also show the fitted EIS results.
Following the reviewers' suggestions, we have revised the relevant content in the manuscript.
5、It would be necessary to give, the scattering (error bars) of all fitted parameters to evaluate the accuracy of the used model.
We sincerely thank the reviewer for careful reading.We investigated combining the real data and the fitted data in the same data display for the EIS, but this would have made the graphic look cluttered.
6、the DFT calculations are strongly recommended.
We sincerely thank the reviewer for careful reading. Quantum chemistry calculations are obtained directly using Gaussian software, so there is no calculation process.
7、Update with new conclusions.
We sincerely thank the reviewer for careful reading. As suggested by the reviewer. We have rewritten the conclusions in the manuscript.
8、Checks all references and correct by the corresponding style of writing references, (journal, journal abbreviation, first and last pages, commas, sources from Web pages are missing a document access date, etc....).
We have re-written this part according to the Reviewer’s suggestion.
Reviewer 3 Report
New approaches for inhibitors are always welcome in the corrosion community, therefore an interesting paper.
A lot of things has to be added/changed in the paperas follows:
- Page 2: what are "Cu monomers"?
- There is no table 2S and 3S
- No icorr, a data
- No EIS data
- No equivalend circuit for the EIS simulation is illustrated and none simulated spectra, too.
- There is no calculation for the inhibitor efficiency shown.
Inhibitor:
The inhibitor ligand and the inhibitor complex are only analysed by IR and XPS and these (alone) are not usefull methods to analyse molecule structures. Other papers use NMR; MS, UV-VIS but at least some combinations of methods to proof the structure (molecular weight).
Here examples:
DOI: 10.2174/1874847301806010001
Author Response
1、Page 2: what are "Cu monomers"?
We were really sorry for our careless mistakes. Thank you for your reminder. It should be Cu atoms.
2、There is no table 2S and 3S
We have revised the relevant content in the manuscript.
3、No icorr, a data
We have revised the relevant content in the manuscript.
4、No EIS data
We have revised the relevant content in the manuscript.
5、No equivalend circuit for the EIS simulation is illustrated and none simulated spectra, too.
We have revised the relevant content in the manuscript.
6、There is no calculation for the inhibitor efficiency shown.
We have revised the relevant content in the manuscript.
7、The inhibitor ligand and the inhibitor complex are only analysed by IR and XPS and these (alone) are not usefull methods to analyse molecule structures. Other papers use NMR; MS, UV-VIS but at least some combinations of methods to proof the structure (molecular weight).
As suggested by the reviewer. We have added 1H-NMR of Thy-2 to the manuscript.
Reviewer 4 Report
The authors prepared bis-thiophene Schiff alkali-copper metal complex for metal corrosion inhibition applications. The following issues must be addressed by the authors to warrant the publication of their work.
1. What is the innovation in this work? Simplified methods on preparing corrosion inhibitors from azole-based compounds and green sources like plants and fruits have been reported in the literature, so what is the improvement here? This has to be thoroughly explained in the introduction.
2. In the introduction, other similar studies should be mentioned in detail for comparison. Authors should also compare the inhibition efficiency of their compounds with others reported in the literature as inhibitors for mild steel in acidic medium.
3. Introduction needs extensive revision. The introduction should be focused on advances done in this field and novelty of the present study.
4. The purity and chemical grade of all reagents used should be stated in the experimental section.
5. Counter electrode surface area should be given in the experimental section. The counter electrode surface area should be way larger than the working electrode surface area. Otherwise, the electrochemical reaction rate is limited by the counter electrode. The half-reaction occurring at the counter electrode can occur faster.
6. The AC perturbation used for EIS should be mentioned in the methodology. The potentiodynamic polarization experiment should mention the scan rate, too.
7. How many experiments were performed at each concentration? This information is very important.
8. The scale bar on SEM/AFM images is difficult to read. Please state in the figure caption instead.
9. Contact angle and SD values should be rounded to the nearest whole numbers.
10. The potential scan rate of 1 mV/s is much larger than the recommended 0.167 mV/s value. The authors should explain this choice.
11. The stabilized OCP values should be presented and compared to Ecorr values obtained from potentiodynamic polarization experiment. The authors should provide the OCP values versus time graph.
12. Some papers that are related to corrosion inhibitors and use of electrochemical techniques for corrosion analysis should be cited. For example: J. Electroanal. Chem. 2021, 880, 114858 and Polym. Int. 2021, 70 (7), 927–937.
Author Response
1、What is the innovation in this work? Simplified methods on preparing corrosion inhibitors from azole-based compounds and green sources like plants and fruits have been reported in the literature, so what is the improvement here? This has to be thoroughly explained in the introduction.
We sincerely thank the reviewer for careful reading. As suggested by the reviewer. We have revised the introduction.
2、In the introduction, other similar studies should be mentioned in detail for comparison. Authors should also compare the inhibition efficiency of their compounds with others reported in the literature as inhibitors for mild steel in acidic medium.
We sincerely thank the reviewer for careful reading. As suggested by the reviewer. We have revised the introduction.
3、Introduction needs extensive revision. The introduction should be focused on advances done in this field and novelty of the present study.
We sincerely thank the reviewer for careful reading. As suggested by the reviewer. We have revised the introduction.
4、The purity and chemical grade of all reagents used should be stated in the experimental section.
We sincerely thank the reviewer for careful reading. As suggested by the reviewer. We have added relevant content to the experimental section.
5、Counter electrode surface area should be given in the experimental section. The counter electrode surface area should be way larger than the working electrode surface area. Otherwise, the electrochemical reaction rate is limited by the counter electrode. The half-reaction occurring at the counter electrode can occur faster.
We sincerely thank the reviewer for careful reading. We use a three-electrode system with platinum wire electrodes as counter electrodes.
6、The AC perturbation used for EIS should be mentioned in the methodology. The potentiodynamic polarization experiment should mention the scan rate, too.
We sincerely thank the reviewer for careful reading. As suggested by the reviewer. We have revised it in the manuscript.
7、How many experiments were performed at each concentration? This information is very important.
We performed three experiments at each concentration and selected the best results.
8、The scale bar on SEM/AFM images is difficult to read. Please state in the figure caption instead.
We have added the relevant content to the legend caption.
9、Contact angle and SD values should be rounded to the nearest whole numbers.
For the contact angle test, we tested three consecutive experiments and finally averaged the results as the result. Due to the small SD values, the experimental results are kept in two places.
10、The potential scan rate of 1 mV/s is much larger than the recommended 0.167 mV/s value. The authors should explain this choice.
We researched similar literature on Schiff bases and settled on the same scan rate of 1 mV/s.
11、The stabilized OCP values should be presented and compared to Ecorr values obtained from potentiodynamic polarization experiment. The authors should provide the OCP values versus time graph.
We think this is an excellent suggestion. We waited for the OCP values to stabilize before starting the test during the experiment, and there was little research on the relationship between COP and time. In future experiments, we will focus on the relationship between OCP values and time.
12、Some papers that are related to corrosion inhibitors and use of electrochemical techniques for corrosion analysis should be cited. For example: J. Electroanal. Chem. 2021, 880, 114858 and Polym. Int. 2021, 70 (7), 927–937.
As suggested by the reviewer. We have cited this document in the manuscript.
Reviewer 5 Report
Dear Authors,
In Figure 3g and 3h, the proposed regression models for Tafel curves based on the coefficient of determination (R2=0.0084 and R2=0.0083, respectively) have practically no functional dependence of the obtained experimental data.
In this case, you need to explain why there is practically no dependence between the proposed regression models and the experimental data or choose new regression models where the coefficients of determination would be >0.85 and which practically best approximate the experimental data.
If you are interested and if you want, you can send me the experimental data from Figure 3g and Figure 3h for Tafel curves and I can suggest regression models that best approximate the experimental data.
Author Response
1、In Figure 3g and 3h, the proposed regression models for Tafel curves based on the coefficient of determination (R2=0.0084 and R2=0.0083, respectively) have practically no functional dependence of the obtained experimental data.
We identified the most suitable model by comparing three adsorption isotherm models.
2、In this case, you need to explain why there is practically no dependence between the proposed regression models and the experimental data or choose new regression models where the coefficients of determination would be >0.85 and which practically best approximate the experimental data.
We identified the most suitable model by comparing three adsorption isotherm models.
3、If you are interested and if you want, you can send me the experimental data from Figure 3g and Figure 3h for Tafel curves and I can suggest regression models that best approximate the experimental data.
We appreciate your assistance and would like to provide you with the necessary information.
Round 2
Reviewer 2 Report
Accept in present form
Author Response
I appreciate your review.
Reviewer 3 Report
Now the presentation of the results are celar.
Some details are missing as follows:
2.1: The sources of the following compunds are missing:
4,6-Diaminopyrimidine
Thiophene-3-carbaldehyde
Acetic Acid
steel sheets
Water (distilled, pure, ultra pure etc.)
DMSO-d6
In figure 1 the designation (A,B,C,...) on the structure is missing and therefore the analysis of the NMR spectrum cannot be reenact.
Please explain in more detail why the EIS data of the inhibitor and the Cu-Inhibitor are simulated with two RC-terms and only one semicircle is visible. It is clear that the is a need for two RC terms (double layer and layer) but it is not visible in the spectra. At least a discussion of the error of the EIS simulation with one or two semicircles are needed.
Author Response
1、Some details are missing as follows:
2.1: The sources of the following compunds are missing:
4,6-Diaminopyrimidine
Thiophene-3-carbaldehyde
Acetic Acid
steel sheets
Water (distilled, pure, ultra pure etc.)
DMSO-d6
We were really sorry for our careless mistake. Thank you for your reminder.
2、In figure 1 the designation (A,B,C,...) on the structure is missing and therefore the analysis of the NMR spectrum cannot be reenact.
We have modified in the figure.
3、Please explain in more detail why the EIS data of the inhibitor and the Cu-Inhibitor are simulated with two RC-terms and only one semicircle is visible. It is clear that the is a need for two RC terms (double layer and layer) but it is not visible in the spectra. At least a discussion of the error of the EIS simulation with one or two semicircles are needed.
We did our best to review the relevant literature but were unable to find a relevant explanation. We appreciate for Reviewers’ warm work earnestly, and hope the correction will meet with approval. Once again, thank you very much for your comments and suggestions.
Reviewer 4 Report
The authors have revised the manuscript according to the comments made. Thus, it can be published in its current format.
Author Response
I appreciate your review.
Reviewer 5 Report
You didn't explain why in Fig. 3g and 3h (now Fig. 7g and 7h), the proposed regression models for Tafel curves based on the coefficient of determination (R2=0.0084 and R2=0.0083, respectively) have practically no functional dependence of the obtained experimental data.
Author Response
We appreciate the reviewers' diligent reading, and with regard to Figures 7g and h, we validate these two Tafel curve regression models using the coefficient of determination in order to establish the best adsorption model for this corrosion inhibitor. These two fit data look to be poor from the graphs, and we will investigate this more in the following research.